# Competent immune responses to SARS-CoV-2 variants in older adults following two doses of mRNA vaccination

Mladen Jergović [1,2,7], Jennifer L. Uhrlaub [1,2,7], Makiko Watanabe[1,2], Christine M. Bradshaw [1,2], Lisa M. White[3], Bonnie J. LaFleur[3], Taylor Edwards [4], Ryan Sprissler[4,5], Michael Worobey[6], Deepta Bhattacharya [1,3] & Janko Nikolich-Žugich [1,2,3 ✉]

Aging is associated with a reduced magnitude of primary immune responses to vaccination. mRNA-based SARS-CoV-2 vaccines have shown efficacy in older adults but virus variant escape is still unclear. Here we analyze humoral and cellular immunity against an early-pandemic viral isolate and compare that to the P.1 (Gamma) and B.1.617.2 (Delta) variants in two cohorts (<50 and >55 age) of mRNA vaccine recipients. We further measure neutralizing antibody titers for B.1.617.1 (Kappa) and B.1.595, with the latter SARS-CoV-2 isolate bearing the spike mutation E484Q. Robust humoral immunity is measured following second vaccination, and older vaccinees manifest cellular immunity comparable to the adult group against early-pandemic SARS-CoV-2 and more recent variants. More specifically, the older cohort has lower neutralizing capacity at 7-14 days following the second dose but equilibrates with the younger cohort after 2-3 months. While long-term vaccination responses remain to be determined, our results implicate vaccine-induced protection in older adults against SARS-CoV-2 variants and inform thinking about boost vaccination.

[1] Department of Immunobiology, University of Arizona College of Medicine, Tucson, AZ, USA. [2] University of Arizona Center on Aging, University of Arizona, College of Medicine, Tucson, AZ, USA. [3] BIO5 Institute, University of Arizona, Tucson, AZ, USA. [4] University of Arizona Genetics Core, University of Arizona, Tucson, AZ, USA. [5] Center for Applied Genetics and Genomic Medicine, University of Arizona, Tucson, AZ, USA. [6] Department of Ecology and Evolutionary Biology, University of Arizona, Tucson, AZ 85721, USA. [7] These authors contributed equally: Mladen Jergović, Jennifer L. Uhrlaub. ✉email: nikolich@arizona.edu

RNA viruses have a high mutation rate resulting in diverse viral populations[1]. Since its emergence in the human population in late 2019, SARS-CoV-2 has infected more than 295 million people globally leading to the appearance of multiple new variants. Several variants of concern (VOC) quickly became dominant in their countries of identification, spreading across the globe. B.1.1.7 (Alpha) was identified in the UK in December 2020 and spread to more than 90 other countries[2]. Other identified VOC with broad spread include B.1.351 (Beta), P.1 (Gamma), and B.1.617.2 (Delta). These lineages are transmitted more efficiently, conferring evolutionary advantage over ancestral virus[3]. Two mRNA vaccines based on the first published genome sequence of SARS-CoV-2 (Wuhan/Hu-1/2019), BNT162b2 (Pfizer) and mRNA-1273 (Moderna), are being deployed to reduce COVID-19 disease severity and transmission, raising the obvious question of whether they will afford similar protection against variants of concern. Reduced plasma antibody (Ab) neutralization titers against pseudovirus in mRNA vaccine recipients has been reported for multiple variants[4]. Another report showed that B.1.351 (Beta) variant might be more effective in escaping humoral immunity than B.1.1.7 (Alpha) variant[5], as one may expect from the location of its mutations to Ab-binding regions. Despite reduced humoral immunity, T cell responses to Alpha and Beta variants were preserved in adult mRNA vaccine recipients[6]. Similar results were obtained with adenovirus vaccine recipients, which showed 3–5 fold lower neutralizing antibody titers against Beta and Gamma variants but uncompromised T cell responses[7].

Efficacy and immunogenicity of many vaccines are known to be decreased in advanced age[8–10]. Fortunately, SARS-CoV-2 mRNA based vaccines were shown to be both well tolerated and highly effective in older adults[11–13]. It remains unknown whether the breadth of the immune response to these mRNA vaccines will remain sufficient for protection against new viral variants.

Here we find that the magnitude and neutralization capacity of humoral memory to these vaccines is not reduced in our older adult cohort (>55 years) as might have been expected based on historic vaccine studies. We confirm lower neutralizing antibody titers against several variants in both of our cohorts. Importantly, robust cellular immunity against WA1, P.1 (Gamma), and B.1.617.2 (Delta) is preserved in most participants. Overall, the effect of age on the immune response to SARS-CoV-2 variants in mRNA vaccinees was measurable, but minimal. The age-related decrease is most evident in the antibody response at 7–14 days after the second vaccine dose. We conclude that effective immunity in older adults is attainable with mRNA vaccines.

## Results and discussion

A total of 40 participants were enrolled in our study before receiving either the BNT162b2 (Pfizer) or mRNA-1273 (Moderna) COVID vaccine. Blood draws were collected prior to initial dose of vaccine, 7–9, and 18–26 days after first vaccine dose, and 7–14 days, 2–3 months, and 6 months after their second dose. These time points are designated as T0, T1, T2, T3, T4, and T5 on the graphs. One participant was excluded from the study because they had a high neutralizing antibody titer and a strong antigen (Ag) specific T cell response before vaccination, indicating prior infection. Our final cohort included 22 participants under 50 years of age (<50) and 17 participants over the age of 55 (>55).

Antibody ELISA assays for the receptor-binding domain (RBD) and S2 region of spike protein on plasma samples at each time point show similar rates and levels of seroconversion between cohorts (Fig. 1a, b). Area under curve (AUC) values confirm that participants over age 55 have robust and comparable antibody responses to the younger cohort with no significant differences measured at any time point (Fig. 1c). Neutralizing antibody test titers utilizing WA1 were also comparable between both cohorts across time points (Fig. 2a). These data taken together are important because they demonstrate that the humoral immune response in older adults is preserved to these mRNA vaccines when tested with the virus they were generated against. We found a diminished capacity of both cohorts to neutralize the P.1(Gamma), AZ-E484Q, and B.1.617.1 (Kappa), and B.1.617.2 (Delta) at the third time point with equivalent neutralization in memory (Fig. 2a and Table 1). E484Q, a mutation in the RBD of spike protein, is present in both B.1.351 (Beta) and P.1 (Gamma) and has already been demonstrated to impact neutralization capacity[14]. We chose AZ-E484Q to test against vaccine induced antibodies because it has spike E484Q and D614G mutations in common with B.1.617.1 (Kappa) but not L452R. L452R has already been shown by others to impact neutralization in pseudotyped virus systems[15] but it remained an open question how E484Q impacts neutralization capacity[16]. Our use of authentic B.1.595 SARS-CoV-2, which in spike bears only E484Q and D614G, demonstrates that E484Q also affords the virus an opportunity to escape neutralization. Understanding how this particular mutation impacts neutralization may be important during the emergence of future variants. Finally, we

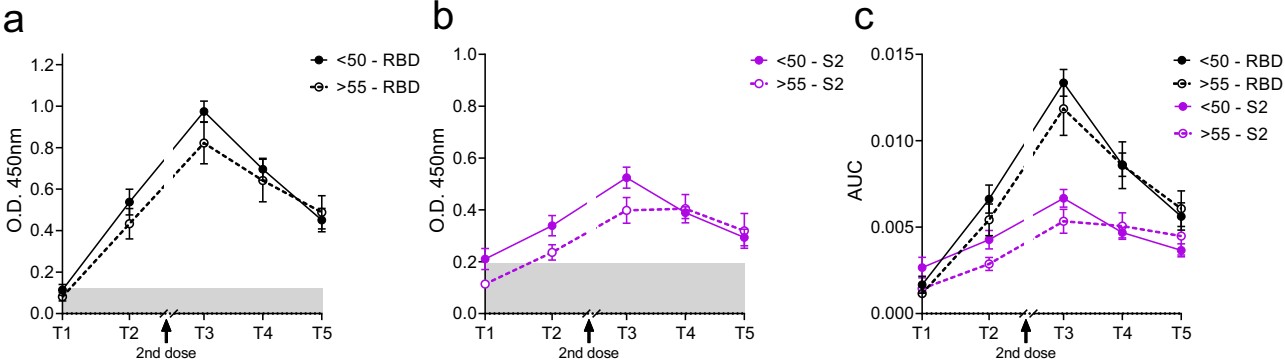

**Fig. 1 Antibody responses to two doses of mRNA vaccine are comparable regardless of age.** Time points denote days 7–9 (T1) and 18–26 (T2) days post first and 7–14 days (T3), 2–3 months (T4), and 6–9 months (T5) post second vaccine dose. $n = 20, 20, 20, 19, 14$ for >55 and 17, 17, 15, 14, 11 for >55 from T1–T5, respectively. Semi-quantitative 1:60 serum dilution ELISA results for reactivity to RBD (**a**) and S2 (**b**) of SARS-CoV-2 spike protein. Gray shaded area indicates positivity threshold for each assay. **c** Quantitative titers for RBD and S2 were calculated for each individual at each time point and are shown as area under the curve (AUC) values. **a**–**c** Two-way ANOVA with post hoc testing for multiple comparisons between cohorts was performed using Sidak's correction. There is no statistical difference between age groups at any time point measured. All data presented as mean values ± SEM.

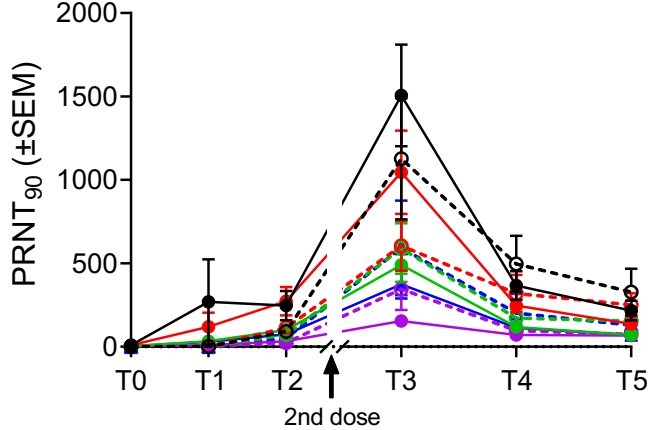

| Symbol | Age | Virus |
|---|---|---|
| ●— | <50 | WA1/2020 |
| -○- | >55 | |
| ●— | <50 | P.1. (Gamma) |
| -○- | >55 | |
| ●— | <50 | B.1.595 (AZ-E484Q) |
| -○- | >55 | |
| ●— | <50 | B.1.617.1 (Kappa) |
| -○- | >55 | |
| ▲— | <50 | B.1.617.2 (Delta) |
| -△- | >55 | |

**Fig. 2 Two doses of mRNA vaccines induce robust neutralization titers in adults under 50 and over 55 that are variably affected, but not abolished, to SARS-CoV-2 variants.** Time points denote pre-vaccination (T0), days 7–9 (T1) and 18–26 (T2) days post first and 7–14 days (T3), 2–3 months (T4), and 6–9 months (T5) post second vaccine dose. $n = 19, 19, 18, 19, 18, 12$ for >55 and 16, 17, 17, 17, 14, 11 for >55 over T0–T5, respectively. Virus neutralization assays were performed using the USA-WA1/2020, P.1 (Gamma), AZ-E484Q, B.1.617.1 (Kappa), and B.1.617.2 (Delta) isolates of SARS-CoV-2. Serial 1:3 dilutions of plasma were assayed for neutralization capacity on Vero cells. The highest dilution capable of preventing >90% of plaques was considered to be the PRNT90 value. Two-sided $P$ values from $t$ test statistics were calculated for pairwise differences using two-way ANOVA. Post hoc testing for multiple comparisons between cohorts and variants was performed using Tukey's multiple comparisons test and determined significant only at time point 3 as represented in Table 1. All data presented as mean values ± SEM.

compared neutralizing antibody titers between vaccine brands at T3 and T4 and determined that antibody responses to the Pfizer vaccine were lower than for Moderna when tested against WA but were statistically indistinguishable when evaluated against the variant viruses (Supplementary Fig. 1A).

Antibody assays are considered the gold-standard when assessing the quality of immunity after vaccination because antibodies can provide sterilizing immunity. However, the establishment of memory B cell populations is also critical to lasting vaccine efficacy as these are the resources available for response to the virus, or a variant, upon next encounter. To assess whether there is an impact of age on the formation of memory B cell populations we measured the frequency and number of circulating memory B cells specific for SARS-CoV-2 pre-vaccination and one week after booster dose by dual staining with tetramers

specific for RBD and S1 using flow cytometry[17] (gating strategy in Fig. 3a). Both groups of participants had a small (0.1–0.3% of all B cells) but detectable population of antigen specific cells double positive for S1 and RBD tetramer (Fig. 3a). This population increased as a percentage of B cells post vaccination in all but two (one from each cohort) participants with no difference between groups (Fig. 3b). By multiplying percentages of tetramer positive B cells with total B cell percentage and lymphocyte counts we have calculated the absolute numbers of SARS-CoV-2 specific B cells in circulation and again observed no difference between cohorts for S1 + RBD + double positive cells (Fig. 3c). Similar results were obtained with S1 single-positive cells again with no difference between adult and older participants post vaccination in percentage of S1 single positive cells (Fig. 3d) or their absolute numbers (Fig. 3e). Next, we examined the phenotype of the SARS-CoV-2 specific B cells (gating strategy in Supplementary Fig. 1B) with respect to class switching, as it has been previously reported that aging is associated with a decline in the percentage and numbers of switched memory B cells[18]. To investigate this possibility, we examined differentiation and class switching of total tetramer (S1+) positive B cells by flow cytometric staining for CD27, IgM, IgD, CD21 and CD11c (representative flow cytometric gating in Supplementary Fig. 1A). Ag-specific B cells from adult and older participants expressed CD27 at identical levels (Supplementary Fig. 1C) and equal numbers of both CD27 positive and negative cells were class switched (Supplementary Fig. 1D, E). Adult and older participants also displayed no difference in classical memory (CD21+) phenotype among the class switched tetramer-positive cells (Supplementary Fig. 1F). Thus overall, we conclude that induction and differentiation of SARS-CoV-2 specific B cells through vaccination was not compromised by aging.

Finally, we measured antigen-specific T cells elicited by vaccinated. Given that ELISpot was previously reported to be a highly sensitive method for detection of rare antigen specific T cells[19], we simultaneously measured the number of T cells expressing costimulatory molecules CD137 and OX-40 by flow cytometry (used in several reports examining SARS-CoV-2 specific T cell immunity -[20,21] (representative flow cytometry in Supplementary Fig. 2A) and performed IFN-γ ELISpot on PBMCs from select participants stimulated with peptide pools corresponding to the spike protein of WA1. Confirming prior data, ELISpot proved to be a much more sensitive method for enumeration of Ag-specific T cells. We detected a statistically significant increase in ELISpots per $10^6$ PBMC's after vaccination analyzing just 5 samples, whereas parallel flow cytometry samples showed no significant differences (Supplementary Fig. 2B–E). The only limitation of ELISpot, as compared to flow cytometry based enumeration, is that total T cell responses are measured without separate quantification of CD4 and CD8 responses.

We also analyzed Ag-specific T cell responses to stimulation with spike protein peptide pools from WA1 versus two VOC by ELISpot. Participant PBMC's were stimulated with 16-mer overlapping peptide pools corresponding to the spike protein of WA1, P.1 (Gamma) and B.1.617.2 (Delta). In accordance with previously published results[22], mRNA vaccines induced a robust T cell response to WA1, the ancestral strain of SARS-CoV-2, which did not differ for Gamma and Delta variants, as evidenced by a tenfold increase in ELISpots from post-vaccination samples stimulated by S peptide pools compared to unstimulated wells (Fig. 4a). Of note, the data represented in Fig. 4a shows concatenated time points for each peptide pool to demonstrate the resolution of this assay. Next, we parsed that data to compare T cell responses of both cohorts at different time points post-vaccination and subtracted the number of spots in the unstimulated wells for each sample to properly calculate the number of

**Table 1 Statistical significance at T3 in Fig. 2.**

|  |  | Summary | Adjusted P Value |
|---|---|---|---|
| WA1/2020 | <50 vs. >55 | ns | 0.2418 |
| P.1 (Gamma) | <50 vs. >55 | ns | 0.0897 |
| B.1.595 (AZ-E484Q) | <50 vs. >55 | ns | 0.9997 |
| B.1.617.1 (Kappa) | <50 vs. >55 | ns | 0.9582 |
| B.1.617.2 (Delta) | <50 vs. >55 | ns | 0.9014 |
| <50 | WA1/2020 vs. P.1 (Gamma) | * | 0.0467 |
| <50 | WA1/2020 vs. B.1.595 (AZ-E484Q) | **** | <0.0001 |
| <50 | WA1/2020 vs. B.1.617.1 (Kappa) | **** | <0.0001 |
| <50 | WA1/2020 vs. B.1.617.2 (Delta) | **** | <0.0001 |
| >55 | WA1/2020 vs. P.1 (Gamma) | * | 0.0226 |
| >55 | WA1/2020 vs. B.1.595 (AZ-E484Q) | * | 0.0157 |
| >55 | WA1/2020 vs. B.1.617.1 (Kappa) | **** | <0.0001 |
| >55 | WA1/2020 vs. B.1.617.2 (Delta) | * | 0.0175 |
| <50 | P.1 (Gamma) vs. B.1.595 (AZ-E484Q) | ** | 0.0049 |
| <50 | P.1 (Gamma) vs. B.1.617.1 (Kappa) | **** | <0.0001 |
| <50 | P.1 (Gamma) vs. B.1.617.2 (Delta) | *** | 0.0002 |

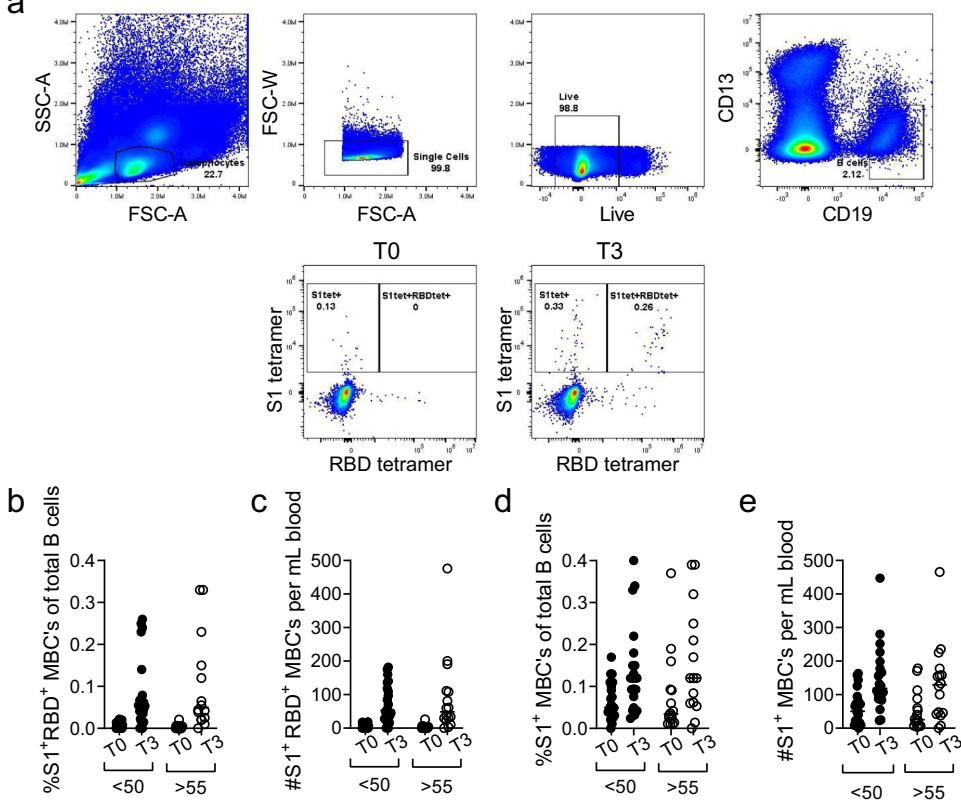

**Fig. 3 Two doses of mRNA vaccine induce SARS-CoV-2 specific memory B cells in adults under 50 and over 55. a** Representative flow cytometry gating. Doublets and dead cells excluded and S1 and RBD tetramer positive cells analyzed as a frequency of total CD19 + B cells; **b** Percentage of tetramer double positive (S1 + RBD + ) B cells were increased after vaccine booster dose (T3) compared to pre-vaccination (T0) and to an equal extent in adult and older adult cohorts. **c** Absolute numbers of S1 + RBD + B cells were also equally increased in <50 and >55 cohorts. **d** Percentage of S1+ tetramer single positive B cells were equally increased in <50 and >55 cohorts; **e** as was their absolute number per ml of blood. $n = 22 < 50$ cohort and $n = 16 > 55$ cohort. Line is median; Mann-Whitney $U$ test.

Ag-specific ELISpots. Data from old mice[23–25] showed that induction of Ag-specific T cell responses becomes delayed and decreased with age. In our data, there was a slightly lower response in the older cohort at day 7 post first dose with all three variants which was statistically significant only with the WA1 peptide pool (Fig. 4b). However, after a second dose both groups had a very robust T cell response against all three viral variants

examined (Fig. 4c–e). Therefore, while we acknowledge that the primary Ag-specific T cell response is lower, and likely delayed, in some of our older participants, we conclude that the outcome after booster dose is competent T cell mediated immunity against WA1 and tested VOC in all mRNA vaccine recipients in our study. This is in agreement with our previous studies of vaccination in aged mice which showed that at least two cycles of

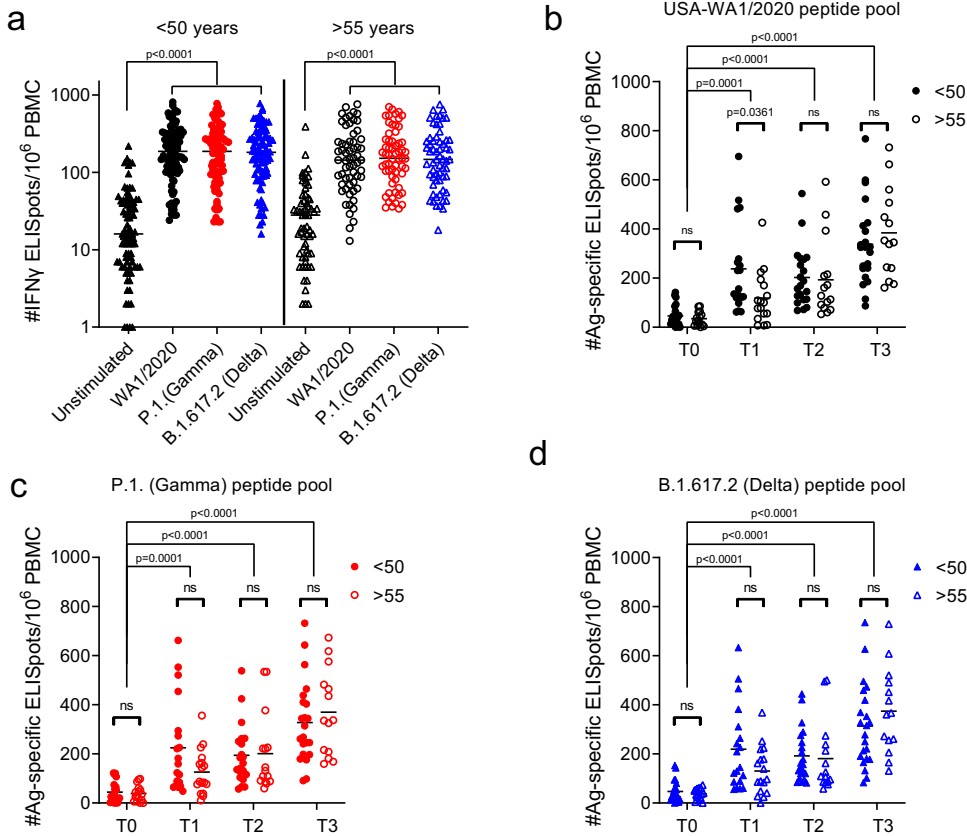

**Fig. 4 Two doses of mRNA vaccines induce SARS-CoV-2 specific T cells reactive to spike peptide pools from all three variants. a** $10^6$ PBMCs per well were stimulated with spike peptide pools from USA-WA1/2020, P.1 (Gamma), and B.1.617.2 (Delta) cultured overnight in pre-coated IFN-γ ELISpot plates. All three peptide pools induced an equally strong IFN-γ response compared to unstimulated wells when all post-vaccination time points (T1, T2, T3) were pooled together. **b** Ag-specific ELISpot numbers calculated by subtracting the unstimulated wells of each participant from the peptide stimulated wells. USA-WA1/2020 induced a lower response in individuals >55 yo after the first dose (T1) and an equal number of ELISpots in <50 and >55 cohorts at later time points (T2 and T3). **c** P.1 (Gamma) peptide pool induced a strong response at all time points post-vaccination which did not differ between <50 and >55 cohorts. **d** Identical results obtained with stimulation with B.1.617.2 (Delta) pool. $n = 22, 19, 21, 22$ for the <50 cohort and $n = 15, 17, 15, 14$ for the >55 cohort across T0, T1, T2, T3, respectively for each. Kruskal Wallis test with Dunn's post hoc correction. Line is Median.

in vivo restimulation are required for adequate ag-specific T cell response in aged animals[26]. All of these data taken together demonstrate the rather expected blunted primary response in older adults and the improvement of this response to the levels seen in adults following a second vaccine dose. Since IFN-γ is not the only effector cytokine produced by T cells following antigen stimulation, we have additionally measured polyfunctional responses. Spike peptide pools induce a dramatic number of IFN-γ spots in comparison to unstimulated wells, but also an increase in IL-2 and GrB spots (Fig. 5a). We observed no difference between the age groups in IL-2 or GrB spots (Fig. 5b) in response to WA/2020 or Delta peptide pools at post-second dose time points (T3 and T4). Similarly, there was no difference between the number of polyfunctional double positive (Fig. 5c) or triple positive cells (Fig. 5d). We also analyzed FLUORISpot responses in recipients of mRNA vaccines from different manufacturers. We did not measure any difference in IFN-γ, IL-2 or GrB T cell responses between recipients of Moderna vs. Pfizer mRNA vaccine (Supplementary Fig. 3A, B).

It is well established that primary immune responses wane with age and contribute to the increased susceptibility to infection experienced by older adults (reviewed in ref. [27]). There is also evidence that generation of immune memory in older adults is reduced, but not maintenance of memory. Studies with multiple pathogens have shown decreased T cell receptor repertoire with age[28,29] which could mean easier escape from existing immunity

for pathogen variants. All of the above-mentioned findings warrant an extensive and long-term monitoring of immunity in SARS-CoV-2 vaccinees, especially those over age 55.

The emergence of SARS-CoV-2 impacted older adults especially hard with more than 80% of deaths in those over age 65[30] and mortality rates rising sharply above the age of 55[31]. Vaccination reduces these rates dramatically[32] demonstrating that the principles of immunology hold true for this virus and that immune memory is what our species is lacking. Well tolerated and effective, SARS-CoV-2 mRNA vaccines induce potent humoral and cellular immune responses[11,22]. Their deployment also offers an opportunity to establish, in a truly immune naive population, correlates, and contours, of protective immunity. Recent elegant studies in rhesus macaques have shown that even sub-sterilizing neutralizing antibody titers are protective and decrease SARS-CoV-2 severity[33]. The same research study demonstrated a protective role for T cell memory responses and this is supported by human research showing that accumulation of oligoclonal CD8 T cells in bronchoalveolar lavage fluid inversely correlated with disease severity[34]. Recently, Collier et al.[35] analyzed 102 partially and 38 fully vaccinated participants and concluded that participants >80 years of age produced lower primary and secondary antibody neutralizing responses, including those against variants[35]. They did not analyze T cell immunity against variants and, somewhat curiously, did not observe increased T cell responses following the second dose in their older

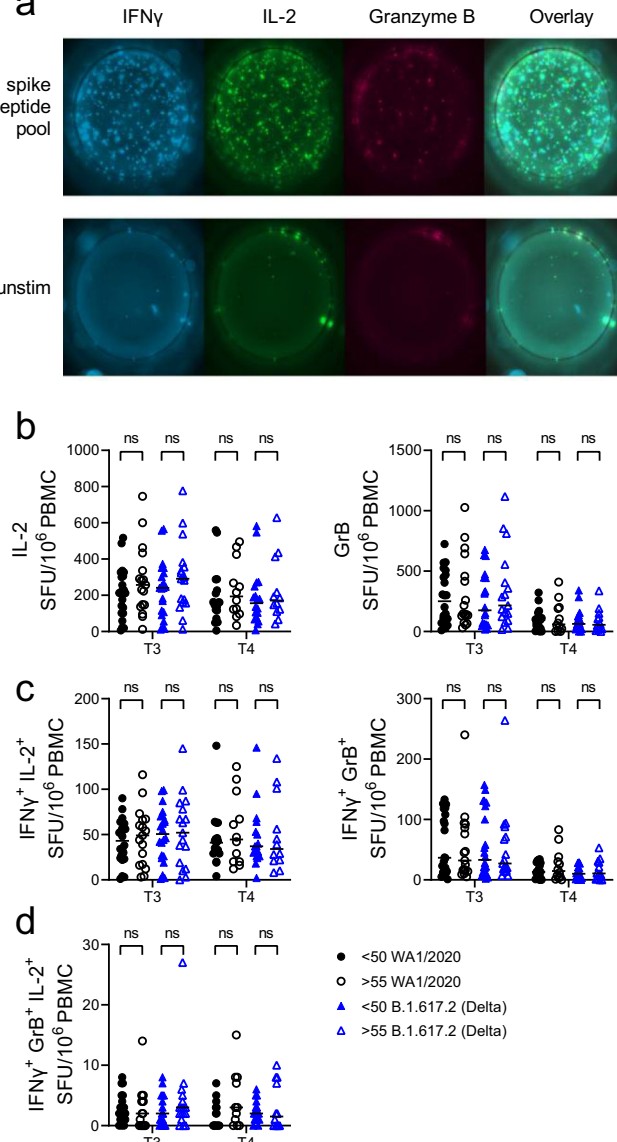

**Fig. 5 mRNA vaccines induce polyfunctional SARS-CoV-2 specific T cells in adult and older adult vaccine recipients. a** $10^6$ PBMCs per well were stimulated with spike peptide pools from USA-WA1/2020 and B.1.617.2 (Delta) cultured overnight in pre-coated IFN-γ, IL-2 and GrB FLUORISpot plates. **b** Ag-specific FLUORISpot numbers were calculated by subtracting the unstimulated wells of each participant from the peptide stimulated wells. USA-WA1/2020 and B.1.617.2 (Delta) pools induced an equal number of IL-2 and GrB spots in <50 and >55 cohorts at each time point. **c** Similarly, numbers of double positive spots (IFN-γ + IL-2+ or IFN-γ + GrB+) did not differ between <50 and >55 cohorts with both peptide pools. **d** Number of triple positive spots (IFN-γ + IL-2+GrB+) were also not different between the age groups and the variant peptide pools. For the <50 cohort, n = 22 at T3 and 17 at T4. For the >55 cohort n = 17 at T3 and 12 at T4. Line is median. Kruskal Wallis test with Dunn's post hoc correction.

groups (>80). These authors argue that older adults—those over 80—remain vulnerable at least until they receive the second vaccine dose. Our results agree with these conclusions with regard to antibody immunity, and suggest that T cell immunity in response to mRNA vaccines is robust in older adults and against variants (Fig. 6), even though we did not analyze participants in the octogenarian bracket. Clinical efficacy of the mRNA vaccines in protecting older adults has been strong, consistent with both

our data and those by ref. [35]. The decrease in antibody titer when challenged with SARS-CoV-2 variants does suggest that the breadth of immunity may be narrower in advanced age; a challenge that can be met with booster doses of heterologous sequence virus. Further studies on the durability and breadth of protection by current and future heterologous vaccines in older adults will be necessary to answer these and other germane questions on their immunity and SARS-CoV-2 protection in older adults.

## Methods

**Study participants**. This study, collection, and use of human blood was approved by the University of Arizona Institutional Review Board (Protocol#2102460536). We collected blood from 23 adults <50 years old and 17 above 55 years old. Demographics are provided in Table 2. Informed consent was obtained for all participants. Participants received a compensation of 25 USD for each blood draw. Samples for all participants were collected prior to vaccination, 1 week after first dose (mRNA vaccine: Pfizer N = 23; Moderna N = 17), day before booster dose and 7–10 days after booster. Many participants were also sampled at 3 or 6–9 months following first dose. The time points are labeled T0, T1, T2, T3, T4, and T5 in all graphs. Blood for complete blood count was collected in BD vacutainer with EDTA and submitted to Sonora Quest Laboratories (Arizona). Blood for peripheral blood mononuclear cells (PBMCs) and plasma was collected in BD Vacutainer with sodium heparin. Plasma was separated by centrifugation at 1000 g for 10 min and PBMC was isolated from the buffy coat by Ficoll-Paque PLUS (GE Healthcare) and cryopreserved in fetal calf serum + 10% DMSO.

**Virus**. SARS-Related Coronavirus 2 (SARS-CoV-2), Isolate USA-WA1/2020, was deposited by Dr. Natalie J. Thornburg at the Centers for Disease Control and Prevention and obtained from the World Reference Center for Emerging Viruses and Arboviruses (WRCEVA). P.1 (Gamma) variant was obtained through BEI Resources, NIAID, NIH: SARS-Related Coronavirus 2, Isolate hCoV-19/Japan/TY7-503/2021, NR-54982, contributed by National Institute of Infectious Diseases. B.1.617.1 was received from BEI (NR-55486 SARS-CoV-2, Isolate hCoV-19/USA/CA-Stanford-15_S02/2021 (Kappa Variant) B.1.617.1G EPI_ISL_1675223). B.1.617.2 was received from WRCEVA. Documentation says it was isolated from a patient at a Galveston (UTMB) hospital; collected 7/6/2021 and passaged once on Vero cells. Strain designation GNL-1205. It has silent mutation C24208T by Sanger sequencing. AZ-E484Q, a B.1.595 virus (GISAID: EPI_ISL_765942) was grown from a nasopharyngeal swab on Calu-3 cells for 48 h. AZ-E484Q has spike D614G and E484Q mutations but has not emerged as a variant of concern. Stocks of SARS-CoV-2 were generated as a single passage from received stock vial (USA-WA1/2020, P.1, B.1.617.1, B.1.617.2) or primary culture (B.1.595) on mycoplasma negative Vero cells (ATCC #CCL-81) at a MOI of 0.005 for 48 h. Supernatant and cell lysate were combined, subjected to a single freeze-thaw, and then centrifuged at 2000 g for 10 min to remove cell debris.

**Plaque reduction neutralization test**. A plaque reduction neutralization test for WA1 and SARS-CoV-2 variants was performed as described in ref. [36]. Briefly, mycoplasma negative Vero cells (ATCC # CCL-81) were plated in flat bottom, 96 well, tissue culture plates and grown overnight. Three-fold serial dilutions (1:20–1:43,740) of plasma/serum samples were incubated with 100 plaque forming units of virus for 1 h at 37 °C. Plasma/serum dilutions plus virus were transferred to the cell plates and incubated for 2 h at 37 °C, 5% $CO_2$ then overlayed with 1% methylcellulose in media. After 96 h, plates were fixed with 10% Neutral Buffered Formalin for 30 min and stained with 1% crystal violet. Plaques were imaged using an ImmunoSpot Versa (Cellular Technology Limited) plate reader. The serum/plasma dilution that contained 10 or less plaques was designated as the NT90 titer.

**ELISpot assays**. T cell specific immunity to peptide pools corresponding to spike, nucleocapsid, and matrix proteins were measured as previously described[17]. Briefly, frozen PBMCs were thawed and rested overnight incubated in 24-well plates overnight at 37 °C with 5% $CO_2$. Following day cells were stimulated in X-VIVO 15 media with 5% male human AB serum containing ~1 nmol of peptide pool corresponding of SARS-CoV-2 Prot S of US-WA1/2020 (ancestral), Gamma (P.1/B1.1.28) and Delta (B.1.617.2) variants or positive control anti-CD3 mAb CD3-2 from or blank media as negative control. Overlapping 16mer peptide pools were purchased from 21st century Biochemicals,Inc. Cell suspensions were transferred to pre-coated Human IFN-γ ELISpot plus kit (Mabtech) and developed after 20 h according to manufacturer instructions. Spots were imaged and counted using an ImmunoSpot Versa (Cellular Technology Limited).

**Antibody ELISA**. Serological assays were performed exactly as in[17,36]. AUC values were calculated using R version 4.0.3 (R Core Team, 2020) and the DescTools package[37].

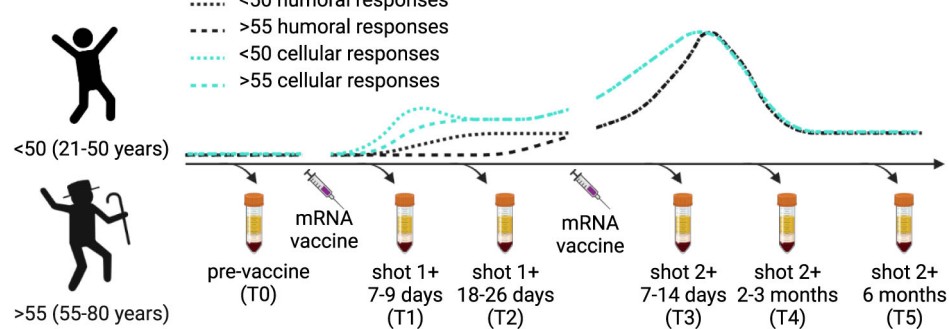

**Fig. 6 Schematic of blood collection time points and overall depiction of T cell and antibody responses measured in adult and older adult cohorts over a two dose series of COVID mRNA vaccine.** Two doses of mRNA vaccine potently stimulate T cell and antibody immune responses, even in older adults, to both ancestral and variant SARS-CoV-2. These data demonstrate that the immune system is quite competent to be leveraged for immune defense and that mRNA vaccines can be an effective strategy.

| Table 2 Demographic characteristics of the cohort. | | |
|---|---|---|
| | **<50 yrs (N = 22)** | **>55 yrs (N = 17)** |
| Gender [N (%)] | | |
| Male | 7 (32%) | 5 (29%) |
| Female | 15 (68%) | 12 (71%) |
| Age [mean (range)] | 31 (20-47) | 67 (56-80) |
| Race [N (%)] | | |
| White | 12 (55%) | 15 (88%) |
| Asian | 6 (27%) | 1 (6%) |
| Unknown | 4 (18%) | 1 (6%) |
| Vaccine Type [N (%)] | | |
| Pfizer | 14 (63%) | 8 (47%) |
| Moderna | 8 (37%) | 9 (53%) |

**Flow cytometry.** Cryopreserved PBMC ($2$–$5 \times 10^6$/sample) were thawed in pre-warmed RPMI-1640 with L-glutamine (Lonza) + 10% FCS. Thawed PBMCS were rested overnight at 37 °C in X-VIVO 15 Serum-free Hematopoietic Cell Medium (Lonza) supplemented with 5% human Ab serum. Cells were stained with surface antibodies in PBS (Lonza) + 2% FCS, and then stained with the live dead fixable blue dye (Thermofisher). B cell tetramers were assembled by mixing 100 µg ml$^{-1}$ of C-terminal AviTagged RBD or S1 (ACROBiosystems) with 100 µg ml$^{-1}$ of streptavidin-PE (eBiosciences) or streptavidin-BV421 (BioLegend), respectively, at a 5:1 molar ratio in which 1/10 of the final volume of streptavidin was added every 5 min. Samples were stained for 1 h at 4 °C. List of antibodies used for flow cytometric staining in Supplementary Table I. Samples were acquired using a Cytek Aurora cytometer (Cytek) and analyzed by FlowJo software (Tree Star).

**FLUORISpot assays.** Cryopreserved PBMC ($5 \times 10^6$/sample) were thawed in prewarmed RPMI-1640 with L-glutamine (Lonza, Basel, Switzerland) + 10% FCS. Thawed PBMCS were rested for 3-4 h at 37 °C in X-VIVO 15 Serum-free Hematopoietic Cell Medium (Lonza) supplemented with 5% human Ab serum. Cells were then stimulated with ~1 nmol of peptide pool corresponding to SARS-CoV-2 spike (S) of US-WA (ancestral), P.1 (Gamma), or B.1.617.2 (Delta). Peptides were 16mer peptide pools, overlapping by 10 amino acids, purchased from 21st century Biochemicals Inc. Cell suspensions were transferred to pre-coated Human IFN-γ, IL-2, Granzyme-B (Gz-B) FLUORISpot kit plates (Mabtech, Inc.) and developed after 48 h according to manufacturer instructions. Spots were imaged and counted using a Mabtech Iris Fluorispot reader (Mabtech).

**Statistical analysis.** SPSS and Graph Pad Prism were used for statistical analysis. Upon inspection of data distribution by Shapiro-Wilks normality test group differences were calculated as described in each figure legend.

**Reporting summary**. Further information on research design is available in the Nature Research Reporting Summary linked to this article.

## Data availability

Source data are provided with this paper. The raw numbers for charts and graphs are available in the Source Data file whenever possible. The flow cytometric data used for analysis of SARS-CoV-2 specific B cells in Fig. 3 has been deposited to flow repository.org and is accessible through accession number FR-FCM-Z56D [https://flowrepository.org/id/FR-FCM-Z56D].

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

## Acknowledgements
Supported in part by the USPHS award R37 AG020719 and the Bowman Professorship in Medical Sciences to J.N.-Z.

## Author contributions
M.J., J.L.U., M.W., D.B., and J.N.Z. designed the study. M.J., J.L.U., M.W., C.M.B., T.E., L.M.W., and R.S. performed the experiments. M.J., J.L.U., and B.J.L. analyzed the data. M.J., J.L.U., and J.N.Z. wrote the manuscript.

## Competing interests
J.N.Ž. is co-chair of the scientific advisory board of and receives research funding from Young Blood Institute, Inc. Sana Biotechnology has licensed intellectual property of D.B. and Washington University in St. Louis. D.B. is a co-founder of Clade Therapeutics. B.J.L. has a financial interest in Cofactor Genomics, Inc. and Iron Horse Dx. The remaining authors declare no competing interests.
