## [Peer Review File · Nature Communications]

REVIEWER COMMENTS

Reviewer #1 (SARS-CoV-2, viral immunity) (Remarks to the Author):

This manuscript compares the age-dependent humoral and cellular immune responses to SARS-CoV-2 mRNA vaccines in individuals >65 years old to individuals <55 years old. They compare neutralizing antibody responses, virus-specific B cell responses, and T cell immune response using ELISpot to SARS-CoV-2 virus, WA1 strain and 2 interesting variants.

Understanding how individuals with advanced age respond to vaccination and how well this will protect against emerging variants of SARS-CoV-2 is a very important topic. The experiments are well designed and controlled. Analysis of both the humoral and cellular immune responses is appreciated.

The main conclusion of the manuscript is that individuals >65 years old had statistically similar humoral and cellular immune responses to individuals <55 years old apart from a small reduction in neutralizing Ab titers against the P.1 variant at the peak of the boosted response. This would be an important conclusion to report. However, it is not clear that this conclusion can be reached due to the extremely limited cohort sizes ($n = 15$ <55 years old) and ($n=9$ >65 years old). As the authors discuss, a larger cohort study (Collier and Ferreira et al) did identify differences in humoral responses in older individuals (<80). In this manuscript, a larger cohort would be needed to conclusively say that no differences truly exist between these groups. Another confounder is that this manuscript includes individuals vaccinated with both the Pfizer and Moderna vaccines and that the percentage of individuals in the <55 year old and >65 year old cohorts receiving these vaccines is not the same. Given it is debated whether the immune response elicited by these vaccines is the same, this further confounds the ability to interpret the results from this small cohort study. Unfortunately, these limitations undermine the ability of the authors to support the primary conclusions of the manuscript.

Reviewer #2 (Infection immunity, vaccine) (Remarks to the Author):

This is a nicely written manuscript describing the immunogenicity of BNT162b2 (Pfizer) and mRNA-1273 (Moderna) vaccines for prevention of SARS-CoV-2 infection in adult <55 yro ($n=15$) and >65yro ($n=9$). The humoral, B and T cellular responses were evaluated, which make this small study interesting. That said, it is still a relatively small cohort with a short observation time, i.e 3 months after second dose. The findings are relevant for the community even if such a short time is evaluated because of the comprehensive analyses. Despite the initial enthusiasm, the analyses of the data raise some question. Throughout the manuscript it seems that similar comparisons are made between the two groups. However, every comparison uses a different statistical method. The small number of samples should require the Kruskal-Wallis non-parametric test across and not the ANOVA, no matter what adjustment is made. The authors should be very specific in the rational for using at least three methods and different adjustments.

It also comes to note that all the comparisons are made using the 24 samples, but in Figure 4A the number of observations (dots) are much higher. The legend does not address what they represent and this should be corrected.

Lastly, not a single reagent utilized to identify the B and T cellular subsets using flow cytometry-based assays is reported, i.e. clones, fluorophores, source, which is not acceptable.

Reviewer #3 (Flavivirus, vaccine) (Remarks to the Author):

The breadth and longevity of the immune response following SARS-CoV-2 infection and vaccination is an area of intense investigation and is important for public health policy and vaccine implementation strategies. Here researchers compared the humoral and cellular immune responses in adult (<55) and old (>65) patients that received the Pfizer or Moderna SARS-CoV-2 vaccine. The authors found little difference between the adult and old immune responses after the full prime-boost vaccine regimen. Humoral immunity from both age groups was diminished against variants of concern yet T cell responses remained high. Overall the findings support the conclusions of this study. However, a recently published study (Collier et al PMID: 34192737) contradict the central conclusion of this study. The limitations in sample size combined with the contradiction to previously published material limits my enthusiasm for this manuscript. Specific critique is outlined below.

Major critique:

The study contradicts previous publications in the field. As noted in the discussion, Collier et al found significantly lower antibody neutralization titers in elderly adults following a prime-boost of the BNT162b2 mRNA vaccine. There are a couple of key differences between the Collier study and the current study. 1.) The Collier study had a larger sample size. 2.) The elderly patient cohort was older in Collier et al (>80). In the current manuscript, there are no individuals over 80. 3.) The Collier cohort only received the BNT126b2 vaccine. In the current manuscript, the study cohort is not segregated by vaccine. However, recent evidence demonstrates that there are discrepancies in the protection against re-infection when comparing individuals that received the Moderna or Pfizer vaccines (<https://www.medrxiv.org/content/10.1101/2021.08.06.21261707v3>). Therefore these patients should be segregated by both age and vaccine. Unfortunately the current study is not sufficiently powered to identify age dependent differences when segregating patient cohort as described. Further in lines 229-231, the authors seem to dismiss the findings of the Collier study as not biologically relevant. The authors do not provide any basis for this statement. And recent guidance from the CDC and the FDA strongly indicates that age-dependent differences in the humoral immune response to BNT162b2 vaccines is very much biologically relevant.

Minor critique:

The 2009 Brien et al study from this same group (and cited herein) demonstrated that both the magnitude and the quality of antiviral T cells is diminished in age. The authors demonstrate in Fig 4 that the magnitude of the antiviral T cell response is equivalent in the young and old cohorts. But have the authors assessed quality. In Fig S2 the authors demonstrate that the IFN γ ELISPOT assay is superior to detect antiviral T cells compared to a flow-based assay (CD137+, OX-40+). However, did the authors quantify polyfunctionality in their analyses.

Figure 2A: PRNT90 titers should be reported on a log scale with a defined Limit of Detection in order to more easily compare all time points.

Figure 4A: What is the source of the patient cells in this figure panel. Not clear in the text and there are many more samples than initially described. Also what time post vaccination are these samples? Are these samples pooled from the different time points? If pooled samples from the same patient, than not appropriate to run stats because dependent variables.

The authors do not sufficiently describe the origin of the S1 and RBD tetramer. The authors cite a MedRXiv preprint. Please include the synthesis details in the M&M.

Figure S1A. Not clear what sub-populations are being analyzed in each plot. Lines/arrows that connect the population in the upper left to the subsequent plot could help.

NCOMMS-21-28260 Competent immune responses to SARS-CoV-2 variants in older adults following mRNA vaccination

We are thankful to all 3 reviewers for their comments on the interest and significance of the work, and for raising constructive criticisms to address. We further thank the editors for the opportunity to revise and improve the manuscript. Our point-by-point reply can be found below.

REVIEWER COMMENTS

Reviewer #1 (SARS-CoV-2, viral immunity) (Remarks to the Author):

1. The main conclusion of the manuscript is that individuals >65 years old had statistically similar humoral and cellular immune responses to individuals <55 years old apart from a small reduction in neutralizing Ab titers against the P.1 variant at the peak of the boosted response. This would be an important conclusion to report. However, it is not clear that this conclusion can be reached due to the extremely limited cohort sizes (n = 15 <55 years old) and (n=9 >65 years old). As the authors discuss, a larger cohort study (Collier and Ferreira et al) did identify differences in humoral responses in older individuals (<80). In this manuscript, a larger cohort would be needed to conclusively say that no differences truly exist between these groups. Another confounder is that this manuscript includes individuals vaccinated with both the Pfizer and Moderna vaccines and that the percentage of individuals in the <55 year old and >65 year old cohorts receiving these vaccines is not the same. Given it is debated whether the immune response elicited by these vaccines is the same, this further confounds the ability to interpret the results from this small cohort study. Unfortunately, these limitations undermine the ability of the authors to support the primary conclusions of the manuscript.

We thank the reviewer for their comments. During the revision period, we have adjusted the age groups in our study to be more in line with recent mortality data demonstrating that those over age 55, much younger than originally reported, are at increased risk of severe COVID (Yanez, N.D., et al. COVID-19 mortality risk for older men and women. *BMC Public Health* **20**, 1742, 2020). We have recruited an additional 7 participants under age 50 and 8 above age 55. Thus, our final sample size is N=40 participants for whom we have pre- and post-mRNA vaccination samples. We feel that this sample size is in line with previously published reports and enables us to perform comprehensive analysis of immune response to vaccination with regards to age and vaccine type. We show that short-term (3 month) responses of T cells following two doses of the vaccine are not influenced by age or mRNA vaccine brand (Figure S3), whereas antibody responses following Pfizer vaccination exhibited lower neutralizing antibody titers compared to Moderna (Figure S1A). While long-term follow ups will be needed to fully understand the longevity of the response, we believe that these results are important since mRNA vaccines do induce a strong response, even in older adults, contrary to what might have been expected based on previously published literature, including ours.

Reviewer #2 (Infection immunity, vaccine) (Remarks to the Author):

1. This is a nicely written manuscript describing the immunogenicity of BNT162b2 (Pfizer) and mRNA-1273 (Moderna) vaccines for prevention of SARS-CoV-2 infection in adult <55 yro (n=15) and >65yro (n=9). The humoral, B and T cellular responses were evaluated, which make

this small study interesting. That said, it is still a relatively small cohort with a short observation time, i.e 3 months after second dose. The findings are relevant for the community even if such a short time is evaluated because of the comprehensive analyses.

We appreciate this reviewer's comment on our study. Please see our response to Reviewer #1 where we have addressed concerns about sample size.

2. Despite the initial enthusiasm, the analyses of the data raise some question. Throughout the manuscript it seems that similar comparisons are made between the two groups. However, every comparison uses a different statistical method. The small number of samples should require the Kruskal-Wallis non-parametric test across and not the ANOVA, no matter what adjustment is made. The authors should be very specific in the rational for using at least three methods and different adjustments.

We apologize for the confusion, and are happy to clarify our methodology. We have used two-way ANOVA in the analyses where immune response was analyzed in relation to two variables (time point and age). Kruskal-Wallis is a non-parametric alternative of one-way ANOVA and therefore not appropriate when a dependent variable is compared in the context of two independent variables. Analyses where immune responses were measured in the context of one variable (such as vaccine brand or age alone, e.g. Figure 3 and Supplemental Figure 3) we have used non-parametric Mann Whitney U test or Kruskal-Wallis ANOVA for multiple comparisons. In all multiple analysis we have used post-hoc correction for multiple comparisons. We feel that this is the most stringent and conservative statistical approach to analyze this data. A revised description of statistical methods can be found between lines 351-356 of the manuscript.

3. It also comes to note that all the comparisons are made using the 24 samples, but in Figure 4A the number of observations (dots) are much higher. The legend does not address what they represent and this should be corrected.

We appreciate this reviewer's call out for the confusion of this figure. We have provided a description in lines 194-199 that these are pooled values from all post-vaccination samples stimulated by S peptide pools compared to unstimulated wells. We apologize for omitting this from the figure legend and we have corrected this in the Figure 4A legend.

4. Lastly, not a single reagent utilized to identify the B and T cellular subsets using flow cytometry-based assays is reported, i.e. clones, fluorophores, source, which is not acceptable.

We fully agree that this important component has to be included as part of the Methods. We have added the reporting summary to the manuscript with all the antibody clones used for flow cytometric analysis; this should have been available to the reviewers as part of the review files.

Reviewer #3 (Flavivirus, vaccine) (Remarks to the Author):

1. Overall the findings support the conclusions of this study. However, a recently published study (Collier et al PMID: 34192737) contradict the central conclusion of this study. The

limitations in sample size combined with the contradiction to previously published material limits my enthusiasm for this manuscript. Specific critique is outlined below.

We thank the reviewer for the comments. Respectfully, we do not agree that our data differs substantially from the Collier study. Both studies measured a decrease in neutralizing antibody titers to SARS-CoV-2 variants. Collier et al. have tested T cell responses only to ancestral (WA/2020) strain and found it decreased in older adults after first vaccine dose but not after the second. Similarly, we have measured decreased IFN- γ after first vaccination dose (Figure 4B) but not after second. We feel that calling this an impaired response to an mRNA vaccine would be an exaggeration since a two-dose regimen was the original standard. We have previously observed similar effects with non-related vaccines in animal models where a second vaccine dose stimulation would produce a competent T cell response equal to younger adult animals. Therefore, the impaired response after the first dose is anticipated in older adults but does not lead to poorer vaccine responses overall because the prescribed vaccine regimen has not been completed. We have added expanded discussion in this regard on manuscript lines 210-214 and 250-256.

2. The study contradicts previous publications in the field. As noted in the discussion, Collier et al found significantly lower antibody neutralization titers in elderly adults following a prime-boost of the BNT162b2 mRNA vaccine. There are a couple of key differences between the Collier study and the current study. 1.) The Collier study had a larger sample size. 2.) The elderly patient cohort was older in Collier et al (>80). In the current manuscript, there are no individuals over 80. 3.) The Collier cohort only received the BNT126b2 vaccine. In the current manuscript, the study cohort is not segregated by vaccine. However, recent evidence demonstrates that there are discrepancies in the protection against re-infection when comparing individuals that received the Moderna or Pfizer vaccines (<https://www.medrxiv.org/content/10.1101/2021.08.06.21261707v3>). Therefore these patients should be segregated by both age and vaccine. Unfortunately the current study is not sufficiently powered to identify age dependent differences when segregating patient cohort as described. Further in lines 229-231, the authors seem to dismiss the findings of the Collier study as not biologically relevant. The authors do not provide any basis for this statement. And recent guidance from the CDC and the FDA strongly indicates that age-dependent differences in the humoral immune response to BNT162b2 vaccines is very much biologically relevant.

In response to reviewer's comments, we have addressed the sample size comment raised by all reviewers in our response to Reviewer #1; the differences (or not) relative to the Collier study under our response to criticism #1 by this reviewer; and the relationship and differences between the two mRNA vaccines in Suppl. Fig. 1 and 3. Moreover, we explicitly acknowledge that we, unlike Collier et al., did not test octogenarians, that as a group may have reacted differently. We have deleted comments on biological differences due to that and other reasons, and have presented what we believe is a more balanced integration of data in the literature (lines 246-258 of the revised manuscript, red font).

Minor critique:

a. The 2009 Brien et al study from this same group (and cited herein) demonstrated that both the magnitude and the quality of antiviral T cells is diminished in age. The authors demonstrate in Fig 4 that the magnitude of the antiviral T cell response is equivalent in the young and old

cohorts. But have the authors assessed quality. In Fig S2 the authors demonstrate that the IFN γ ELISPOT assay is superior to detect antiviral T cells compared to a flow-based assay (CD137+, OX-40+). However, did the authors quantify polyfunctionality in their analyses.

This is an excellent point. We have added Figure 5 in response to this comment providing analysis of polyfunctional T cell responses (IFN- γ , IL-2 and GrB). We found no differences between adults and the older adult group.

b. Figure 2A: PRNT90 titers should be reported on a log scale with a defined Limit of Detection in order to more easily compare all time points.

The PRNT assay is done with three-fold dilutions to provide resolution of biologically relevant differences in titer. For this reason, and since the top of the range is 1:2000, we do not think that a log scale would be appropriate. The limit of detection is inherent within the assay as a zero titer value which means that there was no reduction of viral plaques in the lowest dilution. Therefore, the lower limit of detection is zero.

c. Figure 4A: What is the source of the patient cells in this figure panel. Not clear in the text and there are many more samples than initially described. Also what time post vaccination are these samples? Are these samples pooled from the different time points? If pooled samples from the same patient, than not appropriate to run stats because dependent variables.

We have provided a description in lines 194-199 that these are pooled values from all post-vaccination samples stimulated by S peptide pools compared to unstimulated wells. We apologize for omitting this from the figure legend and we have corrected this in the Figure 4A legend.

d. The authors do not sufficiently describe the origin of the S1 and RBD tetramer. The authors cite a MedRxiv preprint. Please include the synthesis details in the M&M.

We apologize and have added this description to the Materials and Methods lines 337-341.

e. Figure S1A. Not clear what sub-populations are being analyzed in each plot. Lines/arrows that connect the population in the upper left to the subsequent plot could help.

We have taken this reviewers suggestion and added arrows to the subsequent plots in the analysis. This figure is now Figure S1B.

REVIEWERS' COMMENTS

Reviewer #1 (Remarks to the Author):

The authors have adequately addressed all of my comments and concerns.

Reviewer #3 (Remarks to the Author):

Thanks to the authors for addressing the concerns. This manuscript has been improved with the addition of the new data and clarifications. The authors present important data demonstrating the effectiveness of the mRNA vaccines in the older populations. While I agree that these findings are important to the field and should be published, there is one major caveat in the study design that confounds the central conclusion of the study.

The authors show new data that the Pfizer vaccine induces lower neut titers compared to the Moderna vaccine (Fig S1A). The aged and adult cohorts have different frequencies of the Moderna:Pfizer vaccine recipients with the old group skewed towards more Moderna recipients [53% (9/15) vs. 35% (8/23)]. The robust vaccine response in the old could partly be due to a higher percentage of Moderna vaccine recipients in this age group. To address this concern, the authors should segregate their data by vaccine type to include a comparison of adult v. old Moderna recipients and adult v. old Pfizer recipients. These numbers are small, but any broad differences could become apparent.

Minor critique:

Line 246-249. This statement is not accurate. Fig 1F (below) in the Collier et al manuscript demonstrates lower neutralizing antibody titers after the second vaccine dose against the Wuhan and VOC.

Minor, minor critique (Re: Original Point B):

The limit of detection for a viral neut assay is not zero. Your starting dilution is 1:20 therefore it's impossible to detect a PRNT90 of lower than 1:20. But I don't think this really makes a difference in these assays since your titers are much higher here.

Reviewer #3 (Remarks to the Author):

Thanks to the authors for addressing the concerns. This manuscript has been improved with the addition of the new data and clarifications. The authors present important data demonstrating the effectiveness of the mRNA vaccines in the older populations. While I agree that these findings are important to the field and should be published, there is one major caveat in the study design that confounds the central conclusion of the study.

The authors show new data that the Pfizer vaccine induces lower neut titers compared to the Moderna vaccine (Fig S1A). The aged and adult cohorts have different frequencies of the Moderna:Pfizer vaccine recipients with the old group skewed towards more Moderna recipients [53% (9/15) vs. 35% (8/23)]. The robust vaccine response in the old could partly be due to a higher percentage of Moderna vaccine recipients in this age group. To address this concern, the authors should segregate their data by vaccine type to include a comparison of adult v. old Moderna recipients and adult v. old Pfizer recipients. These numbers are small, but any broad differences could become apparent.

We thank the reviewer for these comments. We have now broken Supplemental Figure 1a by vaccine brand and cohort into younger and older adults.

Minor critique:

Line 246-249. This statement is not accurate. Fig 1F (below) in the Collier et al manuscript demonstrates lower neutralizing antibody titers after the second vaccine dose against the Wuhan and VOC.

We thank the reviewer for this comment and we have corrected this mistake now in lines 262-263 of the revised manuscript.

Minor, minor critique (Re: Original Point B):

The limit of detection for a viral neut assay is not zero. Your starting dilution is 1:20 therefore it's impossible to detect a PRNT90 of lower than 1:20. But I don't think this really makes a difference in these assays since your titers are much higher here.